biophysics/biomechanics/biomathematics

bubbles, hydrophobic objects, transport phenomena, cell growth, modelling

**Author for correspondence:**
Kenji Kikuchi
e-mail: kenji@bfsl.mech.tohoku.ac.jp

# Non-biodegradable objects may boost microbial growth in water bodies by harnessing bubbles

Atul Srivastava[1], Kenji Kikuchi[1,2] and Takuji Ishikawa[1,2]

[1]Graduate School of Biomedical Engineering, and [2]Department of Finemechanics, Graduate School of Engineering, Tohoku University, 6-6-01 Aoba, Aramaki, Aoba-ku, Sendai 980-8579, Japan

AS, 0000-0001-5181-4280; KK, 0000-0001-8187-6773; TI, 0000-0002-3573-8414

Given the ubiquity of bubbles and non-biodegradable wastes in aqueous environments, their transport through bubbles should be widely extant in water bodies. In this study, we investigate the effect of bubble-induced waste transport on microbial growth by using yeasts as model microbes and a silicone rubber object as model waste. Noteworthily, this object repeatedly rises and sinks in fluid through fluctuations in bubble-acquired buoyant forces produced by cyclic nucleation, growth and release of bubbles from object's surface. The rise–sink movement of the object gives rise to a strong bulk mixing and an enhanced resuspension of cells from the floor. Such spatially dynamic contaminant inside a nutrient-rich medium also leads to an increment in the total microbe concentration in the fluid. The enhanced concentration is caused by strong nutrient mixing generated by the object's movement which increases the nutrient supply to growing microbes and thereby, prolonging their growth phases. We confirm these findings through a theoretical model for cell concentration and nutrient distribution in fluid medium. The model is based on the continuum hypothesis and it uses the general conservation law which takes an advection–diffusion growth form. We conclude the study with the demonstration of bubble-induced digging of objects from model sand.

## 1. Introduction

Bubbles are ubiquitous. Approximately 10 quintillion ($10^{19}$) bubbles are created every second in water bodies across the planet [1] through various phenomena including sloshing of ocean waves [2], raindrop impacts [3] and sea-floor methanogenesis [4]. These bubbles are vital for the sustenance of aquatic life as they predominantly govern the mixing and distribution of nutrients

and living matter in water bulk [5]. Further, they cause planetary-scale circulation of greenhouse gases (GHGs) by acting as agents of atmosphere–water body interaction [6] and they lead to outbreaks of infectious diseases such as influenza along the shores [7,8]. These positively buoyant bubbles are also nature's transport devices in aqueous environments. A great range of organisms, both macroscopic (insects, snails) [9,10] and microscopic (yeasts) [11], can capture air bubbles on their water-repelling bodies to migrate vertically to the free surface in order to breathe. Such rise is achieved through a reduction in the effective density of the animal via ambient bubble accumulation. In recent years, the naturally rife process of bubble-induced vertical transport has inspired considerable research for a plethora of engineering applications ranging from cargo transportation [12] and environmental remediation [13,14] to electricity generation [15,16] and robot design [17,18].

The bubble-induced transport has two main ingredients: (i) *a bubble generation mechanism* and (ii) *a bubble-trapping object*. Such an object is preferably hydrophobic so that it can easily capture gas bubbles in order to levitate. Various interesting researches in the recent past have played with these two ingredients to have a desired control over the transported object. For instance, Pinchasik *et al.* [19] demonstrated the vertical propulsion of different Janus particles through oxygen bubbles produced by the splitting of hydrogen peroxide. Using similarly generated oxygen bubbles, Song *et al.* [20] devised a pH-responsive cube that can rise up or sink down in an aqueous environment depending upon the acidity of the medium. Not long ago, Luan *et al.* [21] created a photothermally driven hydrophobic motor where they manoeuvred its buoyancy by expanding and shrinking adhered gas bubbles through intermittent light illumination. In another curious work, Zhang *et al.* [16] fabricated a hydrophobic rocket to achieve its cyclic rising and sinking inside a bacterial fermentation system with an intention to generate electricity through Faraday's Law. Furthermore, a lot of studies have employed bubble-induced transport in water filtration to separate hydrophobic microplastics from water and in mineral refining to recover minerals like copper and lead from their ores [22,23].

In this study, we explore the bubble-induced transport of hydrophobic objects with a different objective. According to the report of the United Nations World Water Assessment Programme (UN-WWAP) 2003, about 2 million tons of industrial and agricultural wastes are discharged into the world's waters every day [24]. Given the ubiquity of bubbles, the bubble-induced transport of these wastes should be extremely common in aquatic environments. With this motivation, we set about studying how the phenomenon of bubble-induced transport of non-biodegradable, hydrophobic objects can affect water bodies and the inhabiting microbes. In order to investigate this in a laboratory, we use yeasts as our model microorganism residing inside a nutrient-rich aqueous medium, along with a silicone rubber cuboid as our model waste object. Yeasts are employed as the model microbes because they use major facilitator superfamily as membrane transporters for glucose uptake, which are extensively present in bacteria, archaea and eukaryotes [25]. We show that the model object is transported through bubbles generated by yeast metabolism, giving rise to a stronger bulk mixing and upward suspension of sedimented yeasts (yeasts are denser than water) through fluid wakes generated during its transport. Notably, we report an increase in total yeast concentration in the experimental flask due to the model waste object when compared with the experiments that were done without it. This seemingly counterintuitive result stems from an enhanced nutrient utilization through object's movement within the fluid layer. The mechanism of enhanced cell growth is explained through a continuum model for vertical distribution of yeasts and glucose concentrations in the fluid. The model illustrated that the presence of a dynamic object significantly altered cell and glucose distributions, indicating that the transport of non-biodegradable wastes should influence nutrient circulation and the spread of organic matter in waters. Also, the reported increment in cell concentration by passively dynamic non-biodegradable wastes might be a key observation that can have some implication on the amount of GHGs [26] that are released by microbes at the floors of water bodies and on the payload of nutrients in aquatic ecosystems.

In the next section, we elaborate upon our experimental results, which include describing the phenomenon of bubble-induced vertical transport of a hydrophobic object and the process of sediment suspension through object's wake. In §3, we quantify the enhanced cell transport caused by the motion of the object with the help of a continuum model for vertical distribution of yeasts within the fluid. In §4, we describe how the bubble-induced object transport led to an increase in the total yeast concentration through enhanced nutrient utilization, which is also supported through a mathematical model. Finally, in §5, before presenting our concluding remarks, we show that a rubber object, initially buried underneath a layer of model sand, can make its way into the fluid medium by harnessing bubbles. All experimental techniques, protocols and mathematical methods are described in §6.

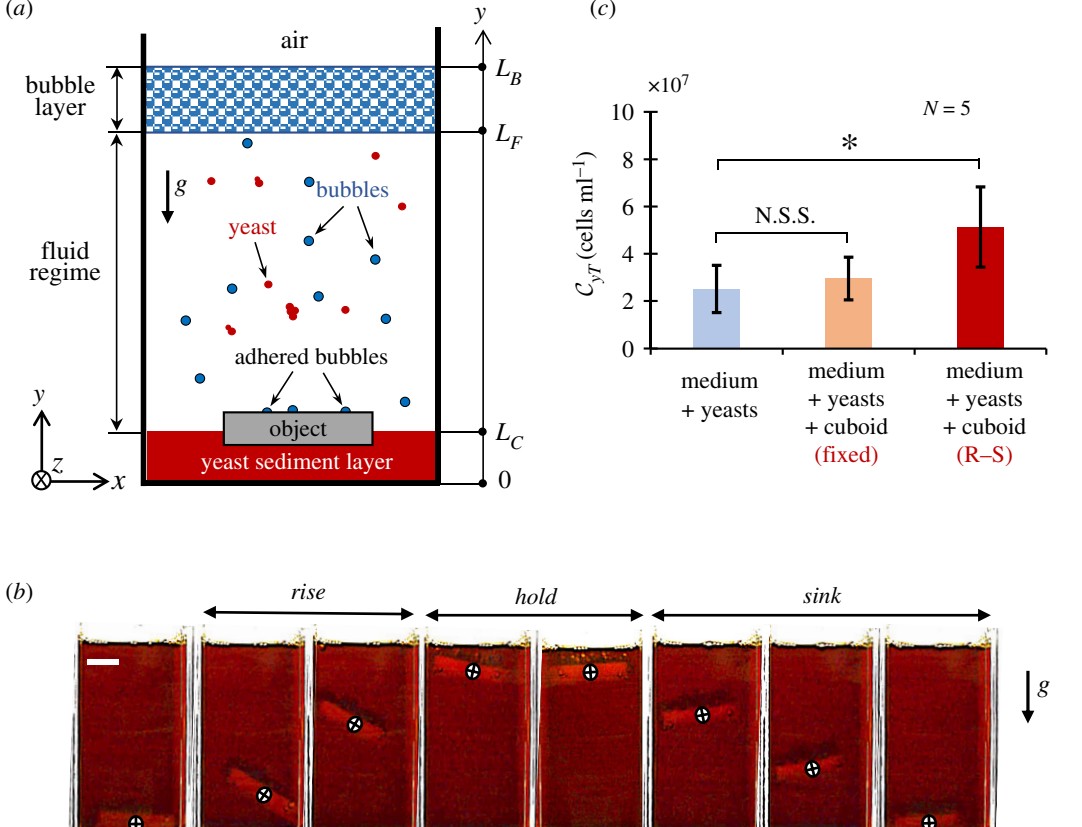

**Figure 1.** Rise–sink (R–S) phenomenon. (*a*) The schematic of an ebullient fluid by yeast fermentation with a foreign object. The *y*-axis is taken vertically upwards, whose origin is at the bottom of the flask. The *x*- and *z*-axes are taken horizontally, as shown in the figure. (*b*) The rising and sinking montage of a silicone cuboid in weakly bubbling fermenting broth. These images are taken for 157 s, 50 h after initial yeast inoculation. Circular crosses trace the orientation and trajectory of the vertically dynamic cuboid. Scale bar is 1 cm. (*c*) The effect of R–S cycle on the total yeast concentration (72 h after initial inoculation). Error bars denote s.d. and asterisk (*) represents $p < 0.05$. N.S.S. stands for no statistical significance. The dimensions of the cuboid are $(\ell_x, \ell_y, \ell_z) = (3, 0.6, 1)$ cm. The significance was determined through *t*-test.

# 2. The rise–sink cycle

The model physical setting for this report is illustrated in figure 1*a* with fermenting yeasts (*Saccharomyces cerevisiae*) as model microorganism residing inside a nutrient broth (standard yeast extract–peptone–dextrose (YPD) medium [27], details can be found in §6.1) containing glucose. As shown, the setting includes three layers, viz: (i) the yeast sediment layer, (ii) the fluid layer and (iii) the bubble layer. The yeast cell layer, at the flask floor, is formed due to multiplication of initially inoculated cells during the early stages of their growth, which is kept together by cell–cell binding stemming from cell surface charges and zymolectin interactions [28]. The nutrient broth or the cell culture fluid constitutes the fluid layer which supplies nutrients for yeast cells to grow. And, lastly, the collective accumulation of $CO_2$ bubbles, released as the consequence of nutrient metabolism by yeasts, creates a bubble layer above the fluid layer. This setting loosely mimics a water body with a sediment layer of live benthic microorganisms, where water with dissolved organic matter constitutes the fluid domain and where bubbles are formed via various phenomenon ranging from wave-breaking to methane generation by cyanobacteria.

For our investigation, to see how the presence of an inert non-biodegradable foreign object affects the microorganisms and their dynamics, we introduce a silicone rubber cuboid ($\rho_b \approx 1.10$ g cm$^{-3}$) within this set-up. As can be seen, the inherent surface roughness and hydrophobicity of the foreign object leads to heterogeneous nucleation and accumulation of gas bubbles on its surface. Depending upon the duration of bubble residence ($t_b = 820.8 \pm 368.0$ s) on the surface and mean bubble departure diameter ($d_b = 1 \pm 0.4$ mm), the density of such object can effectively be reduced below that of the ambient medium. Please note that, in our case, the bubble size distribution on object surface has a mean diameter of 394.7 μm with a standard deviation of 190.9 μm. The further details of bubble behaviour on different

material surfaces, along with a probability distribution for bubble size on silicone rubber, can be found in electronic supplementary material, S1.

After accumulation of yeast-generated bubbles and their sufficiently longer residence on its surface, the effective density of the illustrated rubber cuboid is reduced, triggering its upward movement with a velocity scale of $\mathcal{O}(1)$ cm s$^{-1}$. However, this reduction in density is not long lasting. At the free surface, about a few hundreds of seconds later, the object starts to lose bubbles, either through bubble breakage or via bubble detachment, to lose its bubble-acquired buoyancy. When this loss is enough to overwhelm the density of the ambient fluid, the cuboid sinks down to the floor. The object, then, again starts to gather bubbles to rise and loses bubbles at the top to sink, creating an interesting cycle of rising and sinking. This entire phenomenon is displayed in figure 1b.

The obvious question to arise is what is the effect of object's rise–sink (R–S) cycle on yeast cells? So, we performed three simultaneous experiments which included: (i) a control (YPD broth + yeasts), (ii) fermentation with a fixed object (YPD broth + yeasts + static cuboid), and (iii) fermentation with an object free to rise and sink (YPD broth + yeasts + dynamic cuboid). The fixed-object experiment was done in order to rule out any mysterious chemical effect that silicone rubber might have on yeasts cells and their metabolism. As can be seen from figure 1c, the total yeast concentration $\mathcal{C}_{yT}$ in the flask, measured 72 h post-yeast inoculation, is not significantly different between the control and the fixed-object experiment, whereas it differs significantly with respect to the dynamic-object experiment. This indicates that it is actually the phenomenon of R–S that somehow influences yeast growth and increases the final yeast concentration in the broth. It is probably due to enhanced agitation in fluid due to cyclic rise–sink of the object which raises the area of contact and the relative velocity between cells and the fluid layer [29]. Note that an increase in yeast population through external mechanical mixing has been weakly reported by a few past studies [30,31].

## 2.1. Quasi-natural agitation

Moving on, the fluid agitation due to R–S cycle causes mixing of yeast cells. The yeasts cells already suspended into the fluid layer due to bubbles' wake or yeast–bubble adhesion are influenced by strong fluid disturbances caused by the repeated rising and sinking of a large object in a confined fluid. Such disturbances due to R–S lead these yeast cells to follow chaotic trajectories created within the fluid layer which can be understood as a diffusion phenomenon with an apparent diffusion coefficient $\mathcal{D}_o$, and it works along with the diffusive mixing $\mathcal{D}_b$ caused by bubbles themselves. Our previous work [32] studied, in great detail, such estimation of bubble-induced diffusivity $\mathcal{D}_b$ which is commonly termed as the 'natural mixing'. However, in industrial set-ups (say, breweries), the artificial fluid mixing is often introduced by motor-operated impellers and is generally referred to as 'external/mechanical mixing' [33]. Since, in our case, the mixing occurs due to an immotile foreign object which can be passively transported through naturally generated bubbles, we call such mixing 'quasi-natural'.

The strength of quasi-natural mixing intuitively depends upon the frequency $f_o$ of the R–S cycle. A rapid cycling will cause much stronger agitation when compared with a slower cycle. This is analogous to external mechanical mixing which is a function of the rotational speed of the impeller. We observed that the frequency $f_o$ is dependent upon the bubble flux $f_b$ (i.e. number of bubbles passing through a unit cross-sectional area of the flask in unit time) in an ebullient fluid. Such bubble flux is composed of bubbles that have a mean diameter of $(306.14 \pm 47.5)$ µm, which more or less remain unchanged throughout the course of fermentation (further details in the electronic supplementary material, S1.2). Notice, from figure 2a, $f_o$ is non-zero only when the number of bubbles produced in the fluid is very small $(0 < f_b \leq 0.8$ cm$^{-2}$ s$^{-1})$, which is during first (20–22 h post-inoculation) and last phases (50–52 h post-inoculation) of yeast fermentation process. During moderate to strong bubbling, the amount of gas in the fluid medium is high and the number of bubbles adhering to the object is very large. For this reason, from 22 to 50 h post-inoculation, the effective density of the object remains less than that of the ambient fluid and hence, a zero $f_o$. Note that for determining $f_o$, our counting method involves counting all 'rise events which are followed by a sink event' in a 10 min window. The fortuitous occurrence of R–S only during weak bubbling phases is exploitable by brewing industries, as it can enhance fluid mixing when bubble-induced mixing is feeble and is insufficient to avoid nutrient and temperature inhomogeneities, which are crucial to the quality of the finished product [34].

Another factor that governs the strength of quasi-natural mixing is the size of the object. A large object displaces more fluid during its motion when compared with a smaller one. For the introduced rubber cuboid, we fixed its width $\ell_z$ as half of the flask width $L_z$ i.e. $\ell_z = L_z/2 = 1$ cm. For thickness $\ell_y$, it was set to 0.6 cm, which is liberally larger than the capillary length $\ell_c$ for air–fluid interface as

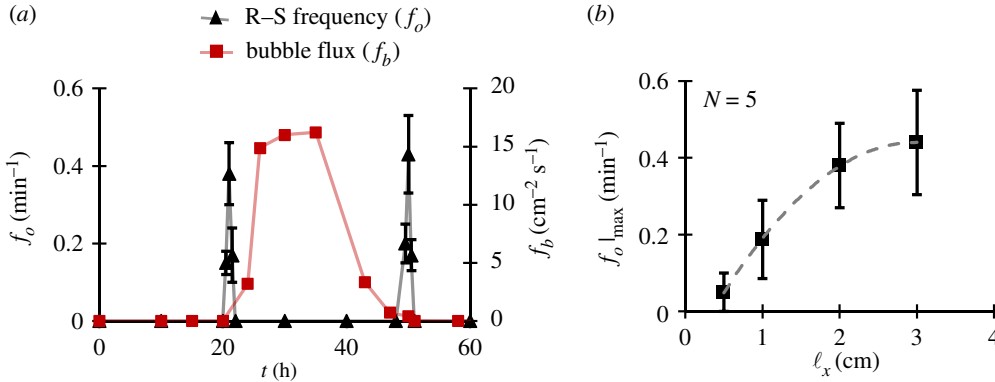

**Figure 2.** Rise–sink (R–S) frequency. (*a*) The long-time record of R–S frequency $f_o$ for a rubber cuboid along with the bubble flux $f_b$ from the flask floor. Note that the rise–sink only occurs during periods of weak bubbling. (*b*) The effect of horizontal length of the rubber cuboid on maximum R–S frequency $f_o|_{max}$ measured 50 h after initial inoculation. Error bars denotes s.d.

$\ell_c = \sqrt{(\gamma/\rho_m g)} \approx 0.27$ cm, where $\gamma$ is the surface tension of the medium and air and $\rho_m$ is the density of the fluid medium. Experimentally speaking, a thinner object is held at the free surface for a very long time (approx. days) due to the relative dominance of surface tension effects over the body weight.

Further, we also performed a size study as in figure 2*b* and found that $f_o$ is dependent on cuboid length $\ell_x$, given $\ell_y$ and $\ell_z$ are unchanged. The graph shows that for smaller lengths, the R–S frequency is lower and as the length is increased $f_o$ grows and then begins to plateau. This tendency possibly occurs due to the stochasticity in bubble-induced buoyant force arising from the randomness in bubble cavity size, bubble detachment diameter and bubble detachment time.

## 2.2. Sediment suspension

Other than quasi-natural mixing, a rising object can also mobilize loosely bounded yeast cells in the sediment layer as illustrated in figure 3*a*. Observe that after gathering enough bubbles, as the cuboid commences its upward journey, it pulls some of the yeast cells with it into the fluid medium through its wake. This is called sediment suspension and it is widely present in aquatic environments. The contents of sediment layer at the bottom of a water body are often suspended due to a variety of mechanisms including methane ebullition [35], bioturbation [36,37], bottom trawling [38] and waves. Such suspension increases the turbidity of the fluid layer reducing the amount of sunlight penetration, which consequently affects the survival of photosynthetic aquatic life. It also has an important implication on cycling of nutrients and vertical distribution of benthic microorganisms [35,39] in water.

In this study, we report a new cause of sediment suspension, i.e. the wake of a large non-biodegradable foreign object which is transported through naturally generated gas bubbles. To visualize the fluid disturbances due to a rising cuboid and its evolution, we performed a two-dimensional particle image velocimetry (PIV) measurement of a part of cuboid's left wake, in a fixed circular window $W$ of radius 3 mm, near the sediment layer. Note that, velocity streamlines form small vortices in the wake (vorticity $\upsilon \approx 40\,s^{-1}$) as in figure 3*b*, which are convected upstream and dissipate rapidly. Corresponding velocity vectors are shown in figure 3*c*.

To characterize whether a given fluid disturbance is strong enough to initiate sediment suspension, Shields parameter $\tau^*$ is generally used [40]. It is a non-dimensional number which is the ratio of fluid force on a particle in the sediment layer and the gravity force, and is given as $\tau^* = \tau/(\rho_y - \rho_m)gd$, where $\tau$ is the shear stress on the sediment layer introduced due to the fluid disturbance, $\rho_y$ is the density of the sediment particle, $\rho_m$ is the density of the fluid layer, $g$ is acceleration due to gravity, and $d$ is the diameter of the sediment particle. In our case, $\rho_y \approx 1.07\,g\,cm^{-3}$, which is the density of a yeast cell, $d \approx 10\,\mu m$ is the mean diameter of a yeast cell, and $\rho_m \approx 1.00\,g\,cm^{-3}$ is the density of the YPD medium. Since for yeast cells, the Reynolds number $Re$ is vanishingly small, the shear stress $\tau$ can be obtained as $\tau = \eta\dot{\gamma}$, where $\eta$ is the viscosity of the medium i.e. 0.001 Pa s, and $\dot{\gamma}$ is the maximum value of the shear strain rate $\partial u_x/\partial y$ in the observed wake, which is about $45\,s^{-1}$. Plugging everything in, we get $\tau^* = 6.55$, implying that the shear stress due to object's wake is strong enough to suspend loosely bounded yeasts from the sediment layer into the fluid layer, as we already demonstrated in figure 3*a*.

Now, to quantify the amount of yeast suspension by the object, we estimated the increment $\delta C_y$ in suspended yeast concentration in the fluid after the object had traversed up. It was done by measuring

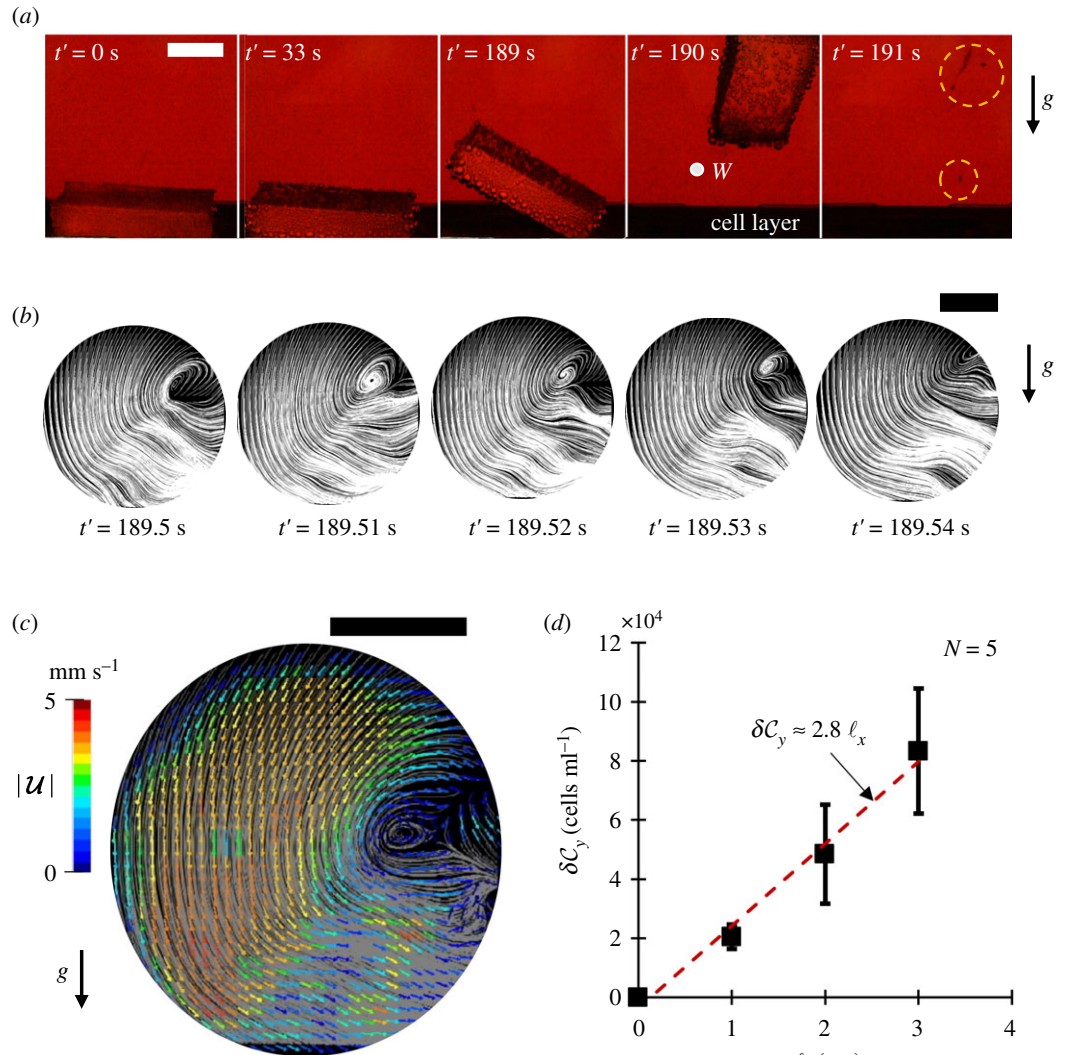

**Figure 3.** Sediment suspension. (a) A sequence of images (scale bar is 1 cm) illustrating that a rising cuboid suspends yeast cells (broken yellow circle) into the medium through its wake. (b) A part of object's wake dynamics with streamlines (scale bar is 1 mm). (c) The particle image velocimetry (PIV) measurement of velocity field in the object's wake in window $W$, where arrows indicate the direction of two-dimensional velocity vectors $\mathcal{U}$ and colours indicate the magnitude $|\mathcal{U}|$ (scale bar is 1 mm). (d) The increment $\delta\mathcal{C}_y$ in yeast cell concentration in fluid regime just after the object has risen up and its variation with respect to the horizontal length of the cuboid. The broken red trend line has $R^2 = 0.95$.

suspended yeast concentration $\delta\mathcal{C}_{y1}$ before the object's rise (which is solely due to yeast–bubble adhesion and bubble-induced yeast suspension), and then, subtracting it from suspended yeast concentration $\delta\mathcal{C}_{y2}$ after the object's rise. Along this line, figure 3d shows how the number of yeasts that are brought into the medium varies with cuboids of different lengths, with the same widths and thicknesses. Note, the increment $\delta\mathcal{C}_y$ grows linearly with the length of the cuboid. Such linearity indicates that the number of yeasts suspended by object's rise is directly proportional to the area of object's surface in contact with the sediment layer when the object is at rest. For this reason, the value of $\delta\mathcal{C}_y$ for curved objects such as a sphere or a cylinder is expected to be small and the corresponding effect on yeasts growth may be difficult to assess. Further, a larger value of $\delta\mathcal{C}_y$ is not only required in brewing industries to have a better nutrient utilization inside a fermenting wort but also a large $\delta\mathcal{C}_y$ is supposed to significantly affect the distribution of organic matter within water bodies.

## 3. Enhanced transport

In the previous section, we elaborated upon our experimental results related to the occurrence of repeated rising and sinking of a silicone rubber object inside a fermenting broth which was divided into two main

**Table 1.** Enhanced transport. (*a*) Enhanced diffusivity by the R–S cuboid. (*b*) Enhanced suspension flux by the R–S cuboid.

| (*a*) diffusivity (cm$^2$ s$^{-1}$) | bubble-induced $\mathcal{D}_b$ | cuboid-induced $\mathcal{D}_o$ |
|---|---|---|
| weak bubbling (*t* = 20–22 h) | $(4.1 \pm 0.7) \times 10^{-3}$ | $(2.3 \pm 0.5) \times 10^{-1}$ at maximum |
| peak bubbling (*t* = 30–35 h) | $(8.2 \pm 0.4) \times 10^{-2}$ at maximum | 0 |
| weak bubbling (*t* = 50–52 h) | $(2.2 \pm 1) \times 10^{-3}$ | $(2.6 \pm 0.6) \times 10^{-1}$ at maximum |
| (*b*) suspension flux (cells cm$^{-2}$ s$^{-1}$) | bubble-induced $\mathcal{J}_b$ | cuboid-induced $\mathcal{J}_o$ |
| weak bubbling (*t* = 20–22 h) | $(2.9 \pm 0.5) \times 10$ | $(3.2 \pm 0.6) \times 10^3$ at maximum |
| peak bubbling (*t* = 30–35 h) | $(5.8 \pm 0.3) \times 10^2$ at maximum | 0 |
| weak bubbling (*t* = 50–52 h) | $(1.6 \pm 0.8) \times 10$ | $(3.6 \pm 0.8) \times 10^3$ at maximum |

subheads. The first one talked about the quasi-natural mixing of the fluid layer by the R–S cycle and the second one highlighted the process of sediment suspension through cuboid's wake. To be more rigorous, in this section, we will try to quantify these findings.

## 3.1. Diffusivity

In §2.1, we talked about the quasi-natural fluid agitation caused by the R–S cycle of a rubber cuboid and mentioned that it can be modelled as a diffusive phenomenon for yeast transport. In the absence of the cuboid, such diffusion occurs due to bubble-induced disturbances, given that the magnitude of fluid agitation occurring due to other small-scale mechanisms such as the Brownian motion and thermal convection are trivially small, approximately $\sim \mathcal{O}(10^{-10})$ cm$^2$ s$^{-1}$, in our case. As discussed in our previous study [32], this bubble-induced diffusivity $\mathcal{D}_b$, in vertical direction, can be estimated through the scaling argument, $\mathcal{D}_b \sim u L_F$, where $u$ is the vertical velocity scale including the effects of bubble rise in the fluid and non-homogeneous distribution of bubbling sources in the yeast sediment layer and $L_F$ is the diffusive length scale which is limited by the height of the fluid layer. In our setting, the maximum value of $\mathcal{D}_b$, when bubbling is at its peak, is $8.2 \times 10^{-2}$ cm$^2$ s$^{-1}$, and during periods of weak bubbling, where R–S occurs, it is approximately $\sim \mathcal{O}(10^{-3})$ cm$^2$ s$^{-1}$. Now, as far as scaling of the enhanced diffusivity $\mathcal{D}_o$ due to the R–S cycle of a cuboid is concerned, its length scale remains limited by the height of the fluid layer $L_F$ and its time scale can be assumed to be the inverse of the R–S frequency $f_o$, yielding $\mathcal{D}_o \sim L_F^2 f_o$, whose maximum value is $2.6 \times 10^{-1}$ cm$^2$ s$^{-1}$, which is about three times the value of maximum bubble-induced diffusivity and is two orders of magnitude larger than the bubble-induced diffusivity in weak bubbling periods. The R–S-induced diffusivity is non-existent during moderate to peak bubbling (table 1*a*).

## 3.2. Upward suspension flux

In §2.2, we reported the suspension of yeast sediment due to a rising cuboid. Without any cuboid, as elaborated in our previous study [32], upward yeast suspension flux $\mathcal{J}_b$ from the sediment layer into the fluid layer occurs due to bubbles' wake and yeast–bubble adhesion as, $\mathcal{J}_b = (N_a + N_w) f_b$, where $N_w$ is the number of yeasts cells brought to the bottom of the fluid medium through bubbles' wake, $N_a$ is the number of cells that are carried by each bubble from the sediment layer into the fluid layer through adhesion (given yeasts are hydrophobic in nature) and $f_b$ is the number of bubbles produced in unit area of the sediment layer per unit time (cm$^{-2}$ s$^{-1}$). However, when an R–S cuboid is present, we have an enhanced suspension flux $\mathcal{J}_o$ due to cuboid's wake, which can be obtained from figure 3*d*, as $\mathcal{J}_o = (\delta \mathcal{C}_y V / A_{xz}) f_o(t)$, where $\delta \mathcal{C}_y$ is the increment yeast concentration inside the fluid due to the cuboid's wake, $V$ is the total volume of the fluid, $A_{xz}$ is the horizontal cross-sectional area of the flask and $f_o$ is the frequency of R–S cycle. From table 1*b*, note that $\mathcal{J}_o$ is about two orders of magnitude higher than $\mathcal{J}_b$ during periods of weak bubbling.

## 3.3. Population distribution model

Moving on, in order to understand how the enhanced transport (i.e. increased diffusivity and increased suspension flux) affects the distribution of yeast concentration within the fluid, it is imperative to develop

a mathematical model. In our case, we want this model to describe the vertical distribution of horizontally averaged yeast concentration. Such model was originally developed in our previous study [32] which employed general conservation of cells by considering advection, diffusion and cell growth. Here, we modify the model to take the effect of R–S cycles into account.

As illustrated in figure 1a, and touched upon in §2, there are three layers in the present setting. Since yeast cells are inoculated near the bottom of the flask (i.e. at $y = 0.5$ cm), after inoculation, all the cells sediment to the floor forming a thin sublayer of yeasts with a typical thickness of one yeast diameter and having a volume fraction $\phi \approx 10^{-4}$. Now, as cells begin to grow, this sublayer starts to get denser. After a certain volume fraction $\phi = \phi_T \approx 0.6$, the layer cannot accommodate any more cells, and hence, the growing cells begin to stack on the top of the tightly packed yeast sublayer. This goes on till we have many tightly packed yeast sublayers forming what we call the yeast sediment layer, which is about 1–2 mm thick. This sediment layer is the source and sink of all yeast cells in the fluid layer and in the bubble layer. We assume that, inside a tightly packed cell layer, advection and diffusion phenomenon can be ignored.

In the fluid layer, the length scale of the bulk motion and the length scale of the yeast concentration distribution are, respectively, large enough compared with the yeast diameter and the typical cell spacing. The length scales for the bulk motion and yeast concentration distribution are of the order of a few centimetres (i.e. height of the free surface). On the other hand, the yeast diameter is approximately 10 μm and the typical cell spacing i.e. $\mathcal{C}_y^{-1/3}$ is $10^{-2}$ cm, during bubbling. The total yeast flux $\mathcal{J}_T$ across the horizontal cross-sectional area, inside the fluid ($L_C < y < L_F$) has an advection–diffusion form, which can be given as (figure 1a)

$$\mathcal{J}_T = N_a f_b \hat{y} - \mathcal{U}_s \mathcal{C}_y \hat{y} - \mathcal{D}\nabla\mathcal{C}_y + V\mathcal{C}_y \tag{3.1}$$

where $\hat{y}$ is a unit vector pointing vertically upwards, $\mathcal{C}_y$ is the yeast concentration averaged over the horizontal cross-section of the flask, $N_a f_b$ is the advective flux due to yeast–bubble adhesion pointing vertically upwards, $\mathcal{U}_s \mathcal{C}_y$ is the advective flux due to yeast sedimentation pointing vertically downwards, where $\mathcal{U}_s$ is the yeast sedimentation velocity and $\mathcal{D}\nabla\mathcal{C}_y$ is the diffusive flux arising due to bubble-induced mixing $\mathcal{D}_b$ and quasi-natural mixing $\mathcal{D}_o$, such that the diffusivity $\mathcal{D}$ is approximated as $\mathcal{D} \approx \mathcal{D}_b + \mathcal{D}_o$. Also, $V\mathcal{C}_y$ is yeast advection due to the fluid velocity disturbances $V$. A similar yeast flux equation can be written for the bubble layer ($L_C < y < L_B$).

If we apply the equation of continuity to equation (3.1), we get

$$\frac{\partial\mathcal{C}_y}{\partial t} = -\nabla.(N_a f_b \hat{y} - \mathcal{U}_s \mathcal{C}_y \hat{y} + V\mathcal{C}_y) + \nabla.(\mathcal{D}\nabla\mathcal{C}_y) + \alpha\mathcal{C}_y \tag{3.2}$$

where $\alpha$ is the yeast growth rate.

Integrating equation (3.2) over the horizontal cross-section of the flask gives us

$$\frac{\partial\mathcal{C}_y}{\partial t} = -\frac{\partial(N_a f_b - \mathcal{U}_s \mathcal{C}_y)}{\partial y} - \mathcal{C}_y \frac{\partial}{\partial y}\left[\iint V.d\mathbf{S}\right] + \mathcal{D}\frac{\partial^2\mathcal{C}_y}{\partial y^2} + \alpha\mathcal{C}_y, \tag{3.3}$$

where $d\mathbf{S} = dS\,y$ is the elemental horizontal cross-sectional area pointing vertically upwards. Now, through conservation of mass $\iint V.d\mathbf{S} = 0$, implying that the fluid velocity disturbances do not contribute to the advection of yeasts (although they cause diffusion). So finally, the governing equation for yeast concentration distribution in vertical direction becomes

$$\frac{\partial\mathcal{C}_y}{\partial t} = -\frac{\partial(N_a f_b - \mathcal{U}_s \mathcal{C}_y)}{\partial y} + \mathcal{D}\frac{\partial^2\mathcal{C}_y}{\partial y^2} + \alpha\mathcal{C}_y, \tag{3.4}$$

where symbols have usual meanings. This equation, conceptually, is similar to the cell conservation equation first proposed by Pedley and co-workers [41,42] to model bioconvection phenomena in dilute cell suspensions of gyrotactic microorganisms.

It is important to mention a couple of points here to avoid confusion. Firstly, since the convection flow and yeast cells satisfy conservation of mass (as we are interested in one-dimensional vertical yeast distribution), fluid velocity disturbances do not contribute to the advective yeast transport, and secondly, in equation (3.4), different transport (advective and diffusive) effects can be superimposed because they have relatively independent origins. Also, the linearity of two kinds of diffusivity can be assumed because $\mathcal{D} \approx \mathcal{D}_o$ during weak bubbling and $\mathcal{D} \approx \mathcal{D}_b$ during moderate to strong bubbling.

For completion, we also need to specify the initial and boundary conditions in the given setting. For the initial condition, since we inoculated yeasts cells near the floor of the flask (i.e. at $y = 0.5$ cm), initial yeast

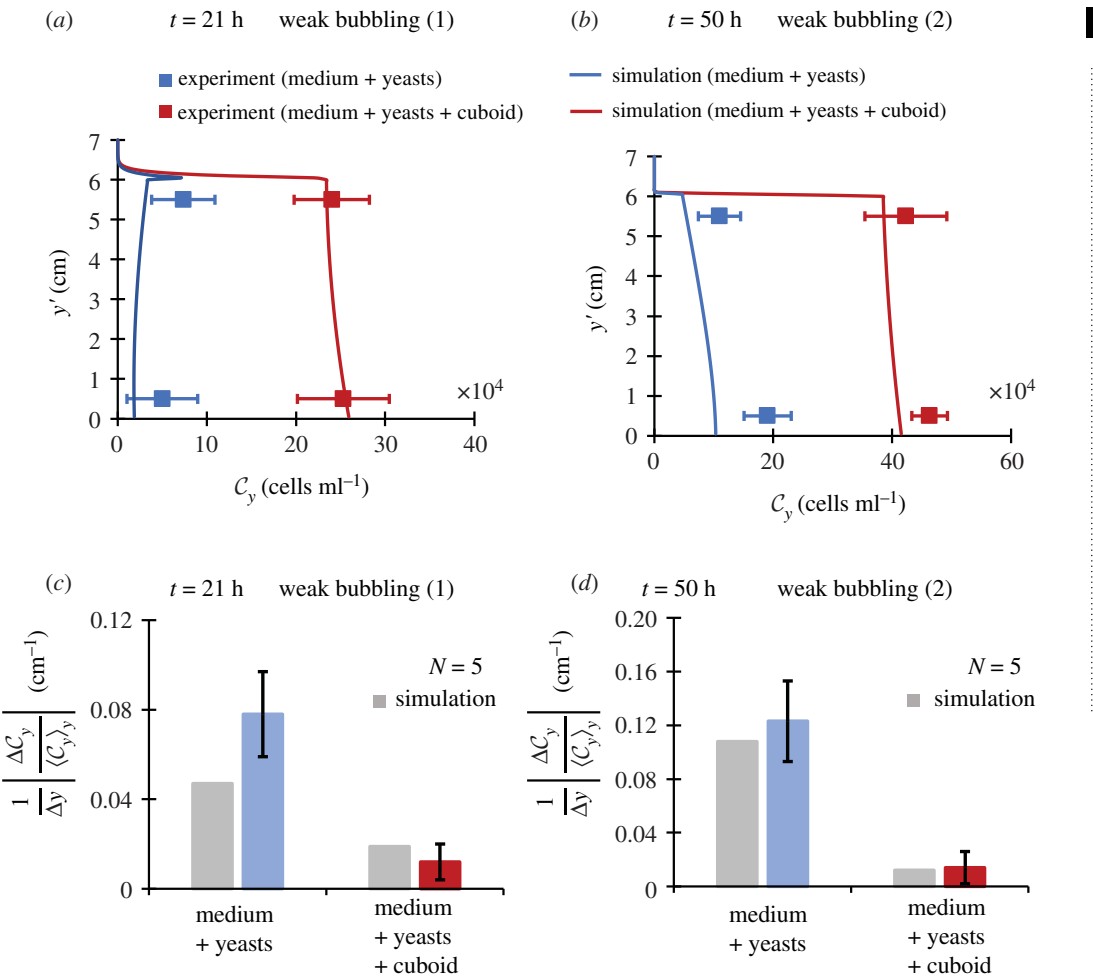

**Figure 4.** Enhanced transport (experiment and the simulation). (*a,b*) The increase in suspended yeast concentration (above the sediment layer i.e. $y' = L_c + y$). (*c,d*) Bar graphs representing the reduction in normalized yeast concentration gradient by a R–S cuboid. All error bars denote s.d. The dimensions of the cuboid are $(\ell_x, \ell_y, \ell_z) = (3, 0.6, 1)$ cm.

concentration is set to $10^5$ cells ml$^{-1}$ for $y \leq 0.5$ cm and zero for $y > 0.5$ cm. At the sediment layer–fluid medium moving interface, the total yeast flux is given by considering upward fluxes due to bubbles and the cuboid and a downward flux due to sedimentation. A similar boundary condition is introduced at the fluid medium–bubble layer interface. At the bubble layer–air interface and the flask floor, we assume there is no flux. The finer details on the mathematical treatment of the cell sediment layer, bubble layer and the boundary conditions can be accessed in the electronic supplementary material S2. The nuances of growth rate $\alpha$ will be explained in the next section as it does not considerably affect the results in this section.

We will now compare the solutions of the governing equation (equation (3.4)) obtained through the method of finite volumes with the experimental measurement of yeast concentration. To do this, we measured the yeast concentration at upper ($y = L_F - 0.5$ cm) and lower ($y = L_c + 0.5$ cm) regions of the fluid layer during two periods of weak bubbling that occur at about 20 and 50 h after initial yeast inoculation. This was performed inside the broth which only had yeast cells and the broth which had yeast cells along with an R–S cuboid. Note from figure 4*a,b* that yeast concentration in the fluid is increased by about 1200% for the first period of weak bubbling and about 300% for second period of weak bubbling. This is due to the phenomenon of sediment suspension from the wake of the rising cuboid as discussed in §2.2. Also, to assess the effect of quasi-natural mixing leading to enhanced diffusivity, we measured the normalized concentration gradient in the fluid layer as in figure 4*c,d*. Note that it reduces by about 85–90% for both periods of weak bubbling and is in good agreement with the simulation data.

From these results, it can be inferred that the presence of a cuboid increases the homogeneity of the fluid layer by curtailing the yeast concentration gradients and increasing the suspended yeasts in the fluid. In the next section, we will show how the discussed enhanced transport by R–S cycles has an influence on yeast growth and nutrient consumption.

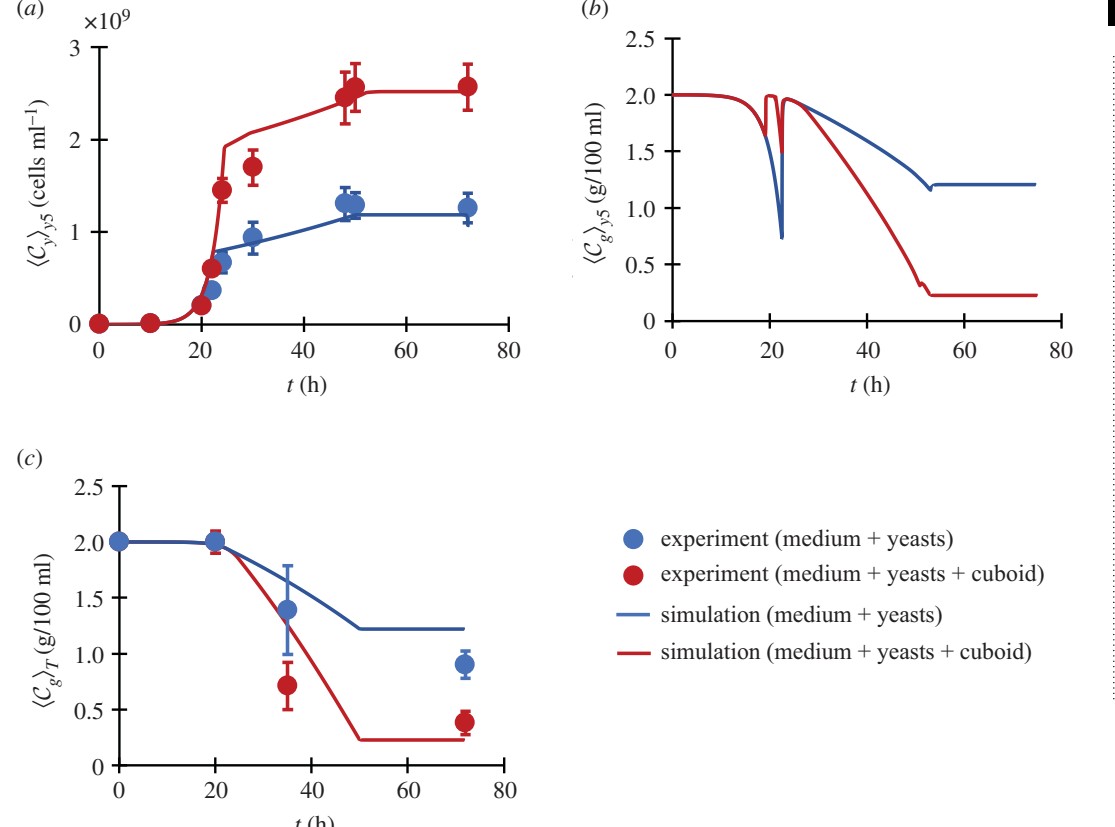

**Figure 5.** Growth and consumption (experiment and simulation). (*a*) The increase in the mean cell concentration near the bottom of the flask. (*b*) The increase in the mean glucose concentration near the bottom of the flask. (*c*) The reduction in the mean glucose concentration in the flask. The dimensions of the cuboid are $(\ell_x, \ell_y, \ell_z) = (3, 0.6, 1)$ cm.

## 4. Enhanced growth and consumption

Recall that in figure 1*c* we reported an increase in total yeast concentration $\mathcal{C}_{yT}$ (fluid layer + sediment layer + bubble layer) within the flask due to the R–S cycles. This can be understood if we couple the yeast population model described above to a simple glucose consumption–diffusion model as

$$\frac{\partial \mathcal{C}_g}{\partial t} = -\xi \mathcal{C}_y + \mathcal{D}_g \frac{\partial^2 \mathcal{C}_g}{\partial y^2}, \tag{4.1}$$

where, $\mathcal{C}_g$ is the glucose concentration with an initial value of 2 g/100 ml, $\xi$ is the glucose consumption parameter approximately equal to $10^{-15}$ g cell$^{-1}$ s$^{-1}$, and $\mathcal{D}_g$ is the diffusivity of glucose. In the fluid layer, without R–S cycles $\mathcal{D}_g \approx \mathcal{D}_b$, whereas, with R–S cycles $\mathcal{D}_g \approx \mathcal{D}_b + \mathcal{D}_o$. Also, inside the packed cell layer and in the bubble layer, glucose diffusion can be neglected. More details in electronic supplementary material, S2.6.

Moving ahead, the growth rate $\alpha$ of yeast cells (cf. equation (3.4)) typically has three phases: (i) a faster exponential phase ($\alpha \approx 0.38$ h$^{-1}$), (ii) a slower exponential phase ($\alpha \approx 0.015$ h$^{-1}$), and (iii) a stationary phase ($\alpha = 0$ h$^{-1}$). Without the R–S cycles, the phase (i) occurs during $t = 0$–20 h post-inoculation, whereas phase (ii) occurs during $t = 20$–50 h and lastly, there is a period of no growth for $t > 50$ h. However, with cuboid R–S cycles, we assume that phase (i) is elongated by a period of 2 h. The effect of glucose supply to the bottom is, therefore, modelled as a corresponding elongation in the 'fast exponential growth phase' by 2 h (i.e. the width of the first R–S frequency spike illustrated in figure 2*a*), which then affects the cell concentration in equation (3.4). During the second spike, the effect of the glucose supply to the floor is negligible as yeasts have already entered the stationary phase.

Observe that in figure 5*a* the yeast concentration $\langle \mathcal{C}_y \rangle_{y5}$ at the bottom of flask ($y \leq 0.5$ cm), obtained through experiments and simulations are in agreement and the effect of R–S cycles on cell growth is visible. The increment in cell concentration $\langle \mathcal{C}_y \rangle_{y5}$ can be understood through an increment in glucose supply $\langle \mathcal{C}_g \rangle_{y5}$ to the bottom during the period of the first R–S spike lasting from 20 to 22 h post-inoculation. Note that the red curve is above the blue one in this period in figure 5*b*. The extra mixing by the R–S cycles during weak

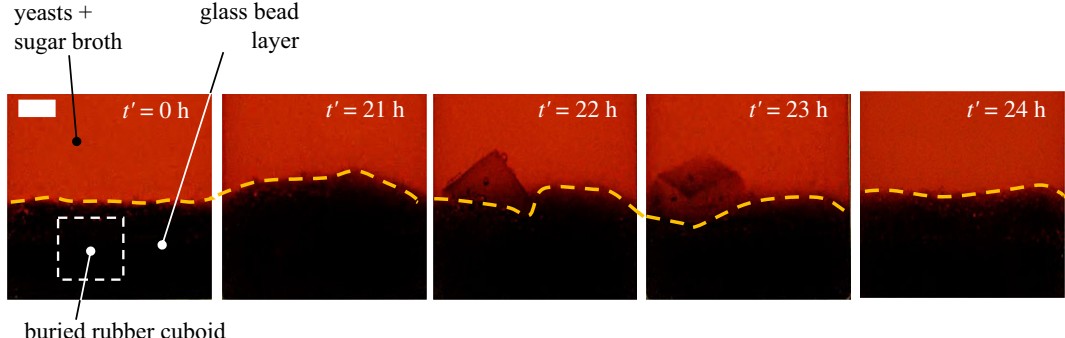

**Figure 6.** Object digging. A cuboid is buried underneath a layer of model sand (glass beads). The diameter of glass beads is between 0.5 and 0.7 mm. Sequential images show the digging of a hydrophobic rubber cuboid by bubbles produced by fermenting yeasts. At $t' = 24$ h, the object has risen up and is at the free surface. Scale bar is 1 cm. The dimensions of the cuboid are $(\ell_x, \ell_y, \ell_z) = (1, 0.6, 1)$ cm.

bubbling significantly increases the glucose concentration at the bottom, using which cells continue their faster exponential phase for a longer time. This tendency can be confirmed in figure 5$c$, in which a reduction in total glucose concentration in the flask $\langle C_g \rangle_T$ is observed both experimentally and numerically.

The results in this section suggest that the reported increase in cell growth by R–S cycles occurs due to an increased supply of glucose to the bottom of the flask where most of the cells are present. This increased supply elongates the faster exponential growth phase for the model microbe, thereby leading to an increment in the cell population.

# 5. Conclusion

In this study, with an objective to understand how bubble-induced movement of non-biodegradable wastes affect water bodies, we started with a model demonstration of cyclic rising and sinking (R–S cycle) of a silicone rubber object in yeast fermentation set up by gas bubbles. Interestingly, this cycle had an effect to increase total cell concentration in the flask. The mechanism of such enhancement in cell growth could be understood through increased nutrient utilization by the cells with the help of R–S cycles. The ingredients to construct the mechanism, i.e. enhanced mixing, enhanced sediment suspension and enhanced glucose consumption, were studied experimentally and a mathematical model was developed which could reproduce the observed phenomena and supported the proposed mechanism.

Since, the bubbles and non-biodegradable wastes are abundant, the demonstrated R–S phenomena should be ubiquitous in water bodies, which should affect the vertical distribution of benthic organic matter, the cycling of nutrients in aqueous media, and the cell growth and nutrient consumption in cell sediments. Moreover, there might be some implications for the aquatic food web and the microbial gas emission by such disturbances in nutrient and microbe distribution. As far as the applicative aspect of this research is concerned, our idea can immediately be exploited to design a fluid mixer for brewing industries that does not require any external power source to operate, but rather harnesses the bubbles produced during fermentation itself.

Although the space of this article is limited, we would like to conclude by reporting a curious consequence of the bubble-induced transport of hydrophobic objects in a more realistic scenario. In figure 6, see that the process of bubble adhesion to a hydrophobic object can not only transport it inside fluid to give rise to R–S cycles as we demonstrated previously, but it can also dig the object out, even if it is buried underneath a thick layer of tightly packed model sand. This, therefore, means that the non-biodegradable wastes that are buried in the floor of water bodies can also influence nutrient circulation in the fluid and thereby, its consumption.

# 6. Material and methods

## 6.1. Cell culture protocol

We used a fermenting strain of yeasts (*S. cerevisiae*) of top-fermenting (Fermentis, SAFALE S-04) kind in this research. The yeast culture was performed in three steps which involved dry yeast hydration, a solid YPDA

(yeast extract, peptone, D-glucose, agar) culture and a liquid YPD culture. In this manuscript, we called the standard laboratory growth medium for yeasts the 'standard YPD medium'. The composition of such medium includes 1% yeast extract, 2% D-glucose and 2% peptone in distilled water. The percentages are given as the weight of the ingredient in grams (g) divided by the volume of distilled water in millilitres (100 ml). We employed yeasts cultured via this procedure for all our investigations. The details of the cell culture protocol can be found in the electronic supplementary material, S3.

## 6.2. Visualization technique

Our experimental technique for observing object transport in figure 1b, sediment suspension in figure 3a and wake observation in figure 3b,c all used the same visualization set-up. The fermenting flask was back-illuminated with a white light source which is an optical fibre or LED and was observed horizontally with the help of a horizontal microscope fitted with a high-speed camera (Phantom v. 7.1, Vision Research) or with a DSLR (D5200, NIKKOR 18–55 mm, Nikon).

## 6.3. Rise–sink frequency $f_o$

To measure the frequency of R–S cycle, we observed the number $\mathcal{N}$ which represents the number of 'rise events that are followed by a sink event', in a 10 min window. The average $f_o$ is then obtained as $f_o = 0.1\,\mathcal{N}$ min$^{-1}$. This was done after different time instants during periods of weak bubbling to obtain figure 2a,b.

## 6.4. Bubble flux $f_b$

It was measured with the help of a DSLR camera (D5200, NIKKOR 18–55 mm, Nikon) focused at the yeast sediment layer and then counting manually the number of bubbling sources and the number of bubbles released by them per second. This was done for entire floor area to obtain the bubble flux as $f_b$ cm$^{-2}$ s$^{-1}$ which is represented in figure 2a and is used to indicate periods of weak bubbling.

## 6.5. Particle image velocimetry

In order to do the PIV measurement of the wake in figure 3b,c, a cell culture flask containing 50 ml of YPD broth was inoculated with $10^5$ yeasts ml$^{-1}$ along with a rubber cuboid. Twenty-one hours after this inoculation (weak bubbling period), a vertical plane within the flask was illuminated from the sides with two red sheet lasers (670 nm) inside a dark chamber [43]. As soon as the cuboid was about to leave the flask floor, high-speed recording was triggered in an observation window of radius 3 mm for 14 s. As the yeast cells scattered enough light, there was no need to seed the fluid layer with baleful tracer beads [44]. Velocity vectors and the streamlines for flow fields were obtained using two-dimensional PIV (Flownizer2D, DITECT) by measuring the cross-correlation between successive frames in movies. The sampling frequency for the PIV was 100 fps with an exposure time of 9.4 ms. The interrogation window size is $25 \times 25$ pixels ($30 \times 30$ µm) and the post-PIV spatial filtering was done using a $3 \times 3$ median filter. Regarding the validation choices, the vector was invalidated if the magnitude of difference of vector and median of the surrounding valid vectors was larger than three pixels. Also, the invalid vectors were replaced by the average of surrounding four valid vectors. Any temporal smoothing for the PIV data was not performed.

## 6.6. Suspended yeast increment $\delta\mathcal{C}_y$

In order to determine the increment in suspended yeast concentration, we counted the number of yeast cells by taking out aliquots of 10 µl from the centre of the fluid layer using a cell counting plate and a vertical microscope (CX-10C, OL-140, HiROX). This was done right before ($\delta\mathcal{C}_{y1}$) and after ($\delta\mathcal{C}_{y2}$) the first rise of the silicone rubber cuboid. As far as the size effect is concerned, we changed the length of the cuboid and kept width and thickness fixed. $\delta\mathcal{C}_y = \delta\mathcal{C}_{y2} - \delta\mathcal{C}_{y1}$ was then calculated to get figure 3d.

## 6.7. Total yeast concentration $\mathcal{C}_{yT}$

The measurement of total yeast concentration ($\mathcal{C}_{yT}$) is slightly different from measurement of fluid yeast concentration ($\mathcal{C}_y$). In the case of $\mathcal{C}_y$, we carefully took out 10 µl aliquots (for cell counting) from the fluid layer so as not to disturb the yeast sediment layer (figure 4a,b). This constituted the suspended yeasts

in the fluid due to various mechanisms including bubbles' wake, object's wake and yeast–bubble adhesion. On the other hand, to obtain $C_{yT}$, we vigorously shook the flask, such that yeast cells from the sediment layer and the bubble layer also got into the fluid. Then, we took out an 10 µl aliquot from the fluid, diluted it $(1:1000)$ and did the cell counting. This was done to get figure 1c.

## 6.8. The bottom yeast concentration $\langle C_y \rangle_{y5}$

The measurement of bottom concentration was done by taking out all the fluid above the height of 5 mm (from the floor) with a stripette. We were careful that it did not touch or disturb the cell layer. After the removal, we were left with a cell layer along with some fluid. We mixed this up vigorously and then diluted it to do the cell counting. This was done to obtain figure 5b and study yeast growth at the bottom.

## 6.9. Normalized concentration gradient $\frac{1}{\Delta y}\left|\frac{(\Delta C_y)}{\langle C_y \rangle_y}\right|$

It was calculated by measuring fluid yeast concentration, through the method described earlier, in upper and lower regions of the fluid. The distance between two points of measurement was $\Delta y = 5$ cm. This gives us the concentration gradient as $\Delta C_y / \Delta y$. However, this is not a true metric to judge the strength of quasi-natural diffusivity, because the object not only causes mixing but also leads to yeast suspension. For this reason, we normalized it with the mean fluid yeast concentration (mean taken along the height) and took its absolute value to obtain figure 4c,d.

## 6.10. Residual glucose measurement $\langle C_g \rangle_T$

To estimate glucose concentration, we used 3,5-dinitrosalicylic acid (DNS) reagent, which measures reducing sugars [45]. For this, we took a 3 ml fluid sample from the flask after shaking it and centrifuged it to remove yeast cells from it. Then, we added 3 ml of freshly prepared DNS reagent (1%) to the sample and heated it at 90°C for about 15 min. Then we added 1 ml of Rochelle salt solution (40%) to the mixture in order to stabilize the colour. It was then cooled to room temperature and the transmittance was measured using a spectrophotometer at 575 nm. The details of reagent preparation, calibration curve and the reason to select 575 nm as our test wavelength can be obtained in the electronic supplementary material, S4.

## 6.11. Growth rate $\alpha$

To measure the growth rate $\alpha$, in all the three phases, we selected a 2 h duration and measured an increase in total yeast concentration $C_{yT}$ within the flask. This was done by choosing various starting times in each phase. With R–S and without R–S, the commencement of stationary phase was after about 50 h post-inoculation, whereas the faster exponential phase lasted for about 20 h without R–S. The end of faster exponential phase with R–S was difficult to measure experimentally, and hence, we assumed it elongates by 2 h, which is equal to the duration of the first R–S spike.

## 6.12. Glucose consumption parameter $\xi$

For estimation of the consumption parameter $\xi$, we measured the increase in cell concentration $\Delta\langle C_y \rangle_{y5}$ along with corresponding decrease in glucose concentration $\Delta\langle C_g \rangle_{y5}$ for a fixed period of time $\Delta t$ near the flask floor. The consumption parameter was then obtained as $\xi = \frac{\Delta\langle C_g \rangle_{y5}}{\Delta\langle C_y \rangle_{y5}\Delta t}$. It was done near the floor and in pre-bubbling phase because all the activity is happening at the floor and there is no mixing present to interfere with the measurements. While yeasts are growing, we assume that the glucose consumption parameter $\xi$ remains constant in both phases of its exponential growth (see, electronic supplementary material, S2.6). However, as soon as yeasts growth hits the stationary phases of no growth (i.e. 50 h post-yeast inoculation), glucose consumption comes to a halt, thereby turning $\xi = 0$ g cell$^{-1}$ s$^{-1}$. This was observed experimentally too.

## 6.13. Numerical methods

We used one-dimensional finite volume method for solving the governing equation (i.e. equation (3.2)) using freely available computation software SCILAB. We exploited QUICK (Quadratic Upstream

Interpolation for Convective Kinematics) scheme for discretization of advection terms and employed second-order central differencing scheme for diffusion term discretization. Time-marching was performed by the explicit Euler method with a time step of 20 s for 80 h.

Data accessibility. The underlying experimental and simulation data can be accessed from the Dryad Digital Repository: https://doi.org/10.5061/dryad.rfj6q579w [46]. The data are provided in electronic supplementary material [47].

Authors' contributions. T.I., K.K. and A.S. formulated and designed the research, A.S. performed simulations and experiments with inputs from T.I. and K.K. All authors were involved in data interpretation and manuscript preparation.

Competing interests. We declare we have no competing interests.

Funding. The authors acknowledge the support of Japanese Society for the Promotion of Science KAKENHI (grants nos. 21H04999, 17H00853, 19H02059) and JST FOREST grant no. JPMJFR2024.

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
