## [Peer Review File · Royal Society Open Science]

Review History

RSOS-210646.R0 (Original submission)

Review form: Reviewer 1

Is the manuscript scientifically sound in its present form?

Yes

Are the interpretations and conclusions justified by the results?

Yes

Is the language acceptable?

Yes

Do you have any ethical concerns with this paper?

No

Have you any concerns about statistical analyses in this paper?

No

Recommendation?

Major revision is needed (please make suggestions in comments)

Comments to the Author(s)

Using a yeast fermentation setup, the authors study, for a model waste particle (rubber), the impact of the bubble-induced waste particle motion. By measuring the yeast cell concentration and the local velocities of the rising/sinking waste particle (which belongs to the relevant results), the authors are able to quantitatively access the enhancement of mixing, sediment suspension and glucose consumption. The results are substantiated by one-dimensional modeling which is in reasonable agreement. The paper is well-written and contains results which are relevant for a variety of systems, from brewery tanks towards marine systems.

I recommend a publication if the following issues are resolved:

p. 4: The numbering (a-c) in the text is somehow misleading because it does not correspond to the (a-c) in Fig.1.

Fig. 3a) and throughout the paper: An information on the size distribution of the bubbles produced, which rise and/or attach to the waste particle is missing.

p. 4 what is a "standard YPD medium?"

p.5, line 23: The term "diffusion coefficient" is not fully correct as it is the combination of advection and diffusion which matters. At least it should be renamed to "apparent diffusion coefficient"

Fig. 2a: The authors show a spike-like behavior of the R-S frequency f_0 . It remains unclear how many measured values are hidden exactly behind this curve. Thus it would be much more transparent if the measured values are given as discrete symbols instead of a solid curve. The same holds true for the bubble flux.

p. 5: The bubble flux f_b is a somehow problematic quantity as it does not take into account the size distribution of the bubbles. In flotation, generally the bubble surface area flux is considered instead. Larger bubbles, upon attachment to the waste particle, are more effective in reducing the effective density.

p.5, line 43/44: Sentence containing "...from 22 h.... the effective density... is less than that of the ambient fluid,... Hence, $f_0=0$ " remains unclear. If the effective density is less than that of the ambient fluid, the object experience buoyancy and should rise. Why f_0 is then zero?

p. 7, line 50: If "c" means a number density, a renaming to N (or n) would be appropriate to avoid confusion with a concentration which typically is given as "c".

p. 8, line 6: " Δn " is here referred to as concentration increment.... Thus, handling of n and c is confusing for me.

p.8, line 35: "...concentration is large compared with yeast diameter and typical cell spacing"?? These quantities have different physical dimensions. How a concentration can be larger than a cell spacing?

It would be good the explain the total yeast flux J_T (eq. 1) in more detail to make its relation to the derivative of n with respect to time (eq. 2) more transparent.

Review form: Reviewer 2

Is the manuscript scientifically sound in its present form?

Yes

Are the interpretations and conclusions justified by the results?

Yes

Is the language acceptable?

Yes

Do you have any ethical concerns with this paper?

No

Have you any concerns about statistical analyses in this paper?

No

Recommendation?

Accept with minor revision (please list in comments)

Comments to the Author(s)

In this paper, the authors report a study on the effect of bubble-induced transport of hydrophobic objects on water bodies and the microbes. They used yeast as a source of bubbles. They found that during the weak bubbling phases with hydrophobic objects, the cell growth was enhanced by the fluid agitation due to the rise-sink cycles. Using a mathematical model, they elucidated the mechanism of the enhancement.

This paper is scientifically rigorous and well-written. In my opinion, the reported results are significant since the considered situation is ubiquitous and the enhancement they found will probably be utilized in industrial fermentation processes, which is not relevant to the judgement in this case.

I think that this paper basically satisfies the criteria for a publication in Royal Society Open Science. So, I recommend the publication of their work after the following minor points are clarified and corrected.

1. If ξ in Eq. 3 is constant, c_g , which must be always positive, can be negative. Is there any dependence on c_g of ξ ?
2. It is unclear how is the cell increase caused by glucose supply to the bottom simulated. Is there any C_g dependence of eq. 2?
3. "The increase" in Fig 5 b is unclear. In the later stage, the red line is below the blue line. More explanation is needed.
4. In page 1, there is the phrase "A great range of animals" but yeast is not an animal. The word "animal" should be corrected.
5. In page 5, 5mixing is probably a misspelling.
6. Fig. 3 (e) in Page 3 of Supplementary document is probably a mistake.

Decision letter (RSOS-210646.R0)

Dear Mr Srivastava

The Editors assigned to your paper RSOS-210646 "Non-biodegradable objects may boost microbial growth in water bodies by harnessing bubbles." have now received comments from reviewers and would like you to revise the paper in accordance with the reviewer comments and any comments from the Editors. Please note this decision does not guarantee eventual acceptance.

Please submit your revised manuscript and required files (see below) no later than 21 days from today's (ie 01-Jun-2021) date. Note: the ScholarOne system will 'lock' if submission of the revision is attempted 21 or more days after the deadline. If you do not think you will be able to meet this deadline please contact the editorial office immediately.

on behalf of Professor Roland Bouffanais (Associate Editor) and Pietro Cicuta (Subject Editor)
openscience@royalsociety.org

Associate Editor Comments to Author (Professor Roland Bouffanais):

This manuscript has been reviewed by two reviewers. The Associate Editor concurs with both reviewers in that this work has some clear merit and provides a conceptual advance in this field. However, one Reviewer has expressed some valid comments and concerns, which will have to be addressed by the authors during the revision stage.

Reviewer comments to Author:

Reviewer: 1

Comments to the Author(s)

Using a yeast fermentation setup, the authors study, for a model waste particle (rubber), the impact of the bubble-induced waste particle motion. By measuring the yeast cell concentration and the local velocities of the rising/sinking waste particle (which belongs to the relevant results), the authors are able to quantitatively access the enhancement of mixing, sediment suspension and glucose consumption. The results are substantiated by one-dimensional modeling which is in reasonable agreement. The paper is well-written and contains results which are relevant for a variety of systems, from brewery tanks towards marine systems.

I recommend a publication if the following issues are resolved:

p. 4: The numbering (a-c) in the text is somehow misleading because it does not correspond to the (a-c) in Fig.1.

Fig. 3a) and throughout the paper: An information on the size distribution of the bubbles produced, which rise and/or attach to the waste particle is missing.

p. 4 what is a "standard YPD medium?"

p.5, line 23: The term "diffusion coefficient" is not fully correct as it is the combination of advection and diffusion which matters. At least it should be renamed to "apparent diffusion coefficient"

Fig. 2a: The authors show a spike-like behavior of the R-S frequency f_0 . It remains unclear how many measured values are hidden exactly behind this curve. Thus it would be much more transparent if the measured values are given as discrete symbols instead of a solid curve. The same holds true for the bubble flux.

p. 5: The bubble flux f_b is a somehow problematic quantity as it does not take into account the size distribution of the bubbles. In flotation, generally the bubble surface area flux is considered instead. Larger bubbles, upon attachment to the waste particle, are more effective in reducing the effective density.

p.5, line 43/44: Sentence containing "...from 22 h.... the effective density... is less than that of the ambient fluid,.... Hence, $f_0=0$ " remains unclear. If the effective density is less than that of the ambient fluid, the object experience buoyancy and should rise. Why f_0 is then zero?

p. 7, line 50: If "c" means a number density, a renaming to N (or n) would be appropriate to avoid confusion with a concentration which typically is given as "c".

p. 8, line 6: " Δn " is here referred to as concentration increment.... Thus, handling of n and c is confusing for me.

p.8, line 35: "...concentration is large compared with yeast diameter and typical cell spacing"?? These quantities have different physical dimensions. How a concentration can be larger than a cell spacing?

It would be good the explain the total yeast flux J_T (eq. 1) in more detail to make its relation to the derivative of n with respect to time (eq. 2) more transparent.

Reviewer: 2

Comments to the Author(s)

In this paper, the authors report a study on the effect of bubble-induced transport of hydrophobic objects on water bodies and the microbes. They used yeast as a source of bubbles. They found that during the weak bubbling phases with hydrophobic objects, the cell growth was enhanced by the fluid agitation due to the rise-sink cycles. Using a mathematical model, they elucidated the mechanism of the enhancement.

This paper is scientifically rigorous and well-written. In my opinion, the reported results are significant since the considered situation is ubiquitous and the enhancement they found will probably be utilized in industrial fermentation processes, which is not relevant to the judgement in this case.

I think that this paper basically satisfies the criteria for a publication in Royal Society Open Science. So, I recommend the publication of their work after the following minor points are clarified and corrected.

1. If ξ in Eq. 3 is constant, c_g , which must be always positive, can be negative. Is there any dependence on c_g of ξ ?
2. It is unclear how is the cell increase caused by glucose supply to the bottom simulated. Is there any C_g dependence of eq. 2?
3. "The increase" in Fig 5 b is unclear. In the later stage, the red line is below the blue line. More explanation is needed.
4. In page 1, there is the phrase "A great range of animals" but yeast is not an animal. The word "animal" should be corrected.
5. In page 5, 5mixing is probably a misspelling.
6. Fig. 3 (e) in Page 3 of Supplementary document is probably a mistake.

===PREPARING YOUR MANUSCRIPT===

If you have been asked to revise the written English in your submission as a condition of publication, you must do so, and you are expected to provide evidence that you have received language editing support. The journal would prefer that you use a professional language editing service and provide a certificate of editing, but a signed letter from a colleague who is a native

speaker of English is acceptable. Note the journal has arranged a number of discounts for authors using professional language editing services (<https://royalsociety.org/journals/authors/benefits/language-editing/>).

===PREPARING YOUR REVISION IN SCHOLARONE===

<https://royalsociety.org/journals/authors/author-guidelines/#supplementary-material> to include a suitable title and informative caption. An example of appropriate titling and captioning may be found at https://figshare.com/articles/Table_S2_from_Is_there_a_trade-

off_between_peak_performance_and_performance_breadth_across_temperatures_for_aerobic_sc
ope_in_teleost_fishes_/3843624.

Author's Response to Decision Letter for (RSOS-210646.R0)

See Appendix A.

RSOS-210646.R1 (Revision)

Review form: Reviewer 1

Is the manuscript scientifically sound in its present form?

Yes

Are the interpretations and conclusions justified by the results?

Yes

Is the language acceptable?

Yes

Do you have any ethical concerns with this paper?

No

Have you any concerns about statistical analyses in this paper?

No

Recommendation?

Accept with minor revision (please list in comments)

Comments to the Author(s)

The authors have incorporated my criticism in an appropriate manner. I recommend the publication of the manuscript after correcting the following minor issues:

-) p.13, line: 18: ...“includes three major regimes, viz: (i) the yeast sediment layer, (ii) the fluid layer, and (iii) the bubble layer“: Are „regimes“ the most suited description for the layers (i-ii)?

-) An explanation of the term “standard YPD medium“, although discussed in the response, has not found entrance into the manuscript.

-) p.18, line 12: „where is the yeast sedimentation velocity“: Is the "y" a typo? Should it not read only „U_s is the velocity...“

Review form: Reviewer 2

Is the manuscript scientifically sound in its present form?

Yes

Are the interpretations and conclusions justified by the results?

Yes

Is the language acceptable?

Yes

Do you have any ethical concerns with this paper?

No

Have you any concerns about statistical analyses in this paper?

No

Recommendation?

Accept as is

Comments to the Author(s)

The manuscript has been well improved, and I think the referees' comments have been addressed. In my opinion, the manuscript is ready for a publication.

Decision letter (RSOS-210646.R1)

Dear Mr Srivastava

On behalf of the Editors, we are pleased to inform you that your Manuscript RSOS-210646.R1 "Non-biodegradable objects may boost microbial growth in water bodies by harnessing bubbles." has been accepted for publication in Royal Society Open Science subject to minor revision in accordance with the referees' reports. Please find the referees' comments along with any feedback from the Editors below my signature.

Please submit your revised manuscript and required files (see below) no later than 7 days from today's (ie 09-Aug-2021) date. Note: the ScholarOne system will 'lock' if submission of the revision is attempted 7 or more days after the deadline. If you do not think you will be able to meet this deadline please contact the editorial office immediately.

Please note article processing charges apply to papers accepted for publication in Royal Society Open Science (<https://royalsocietypublishing.org/rsos/charges>). Charges will also apply to

papers transferred to the journal from other Royal Society Publishing journals, as well as papers submitted as part of our collaboration with the Royal Society of Chemistry (<https://royalsocietypublishing.org/rsos/chemistry>). Fee waivers are available but must be requested when you submit your revision (<https://royalsocietypublishing.org/rsos/waivers>).

on behalf of Professor Roland Bouffanais (Associate Editor) and Pietro Cicuta (Subject Editor)
openscience@royalsociety.org

Associate Editor Comments to Author (Professor Roland Bouffanais):

Comments to the Author:

I concur with the Reviewers in that the manuscript has been greatly improved after this first round of revision. However, one reviewer still recommends some further edits, which should easily be implementable.

Reviewer comments to Author:

Reviewer: 2

Comments to the Author(s)

The manuscript has been well improved, and I think the referees' comments have been addressed. In my opinion, the manuscript is ready for a publication.

Reviewer: 1

Comments to the Author(s)

The authors have incorporated my criticism in an appropriate manner. I recommend the publication of the manuscript after correcting the following minor issues:

-) p.13, line: 18: ...“includes three major regimes, viz: (i) the yeast sediment layer, (ii) the fluid layer, and (iii) the bubble layer“: Are „regimes“ the most suited description for the layers (i-ii)?

-) An explanation of the term “standard YPD medium”, although discussed in the response, has not found entrance into the manuscript.

-) p.18, line 12: „where U_s is the yeast sedimentation velocity“: Is the "y" a typo? Should it not read only „ U_s is the velocity...“

===PREPARING YOUR MANUSCRIPT===

===PREPARING YOUR REVISION IN SCHOLARONE===

- If you are requesting a discretionary waiver for the article processing charge, the waiver form must be included at this step.
- If you are providing image files for potential cover images, please upload these at this step, and inform the editorial office you have done so. You must hold the copyright to any image provided.
- A copy of your point-by-point response to referees and Editors. This will expedite the preparation of your proof.

- Ensure that your data access statement meets the requirements at <https://royalsociety.org/journals/authors/author-guidelines/#data>. You should ensure that you cite the dataset in your reference list. If you have deposited data etc in the Dryad repository, please only include the 'For publication' link at this stage. You should remove the 'For review' link.
- If you are requesting an article processing charge waiver, you must select the relevant waiver option (if requesting a discretionary waiver, the form should have been uploaded at Step 3 'File upload' above).
- If you have uploaded ESM files, please ensure you follow the guidance at <https://royalsociety.org/journals/authors/author-guidelines/#supplementary-material> to include a suitable title and informative caption. An example of appropriate titling and captioning may be found at https://figshare.com/articles/Table_S2_from_Is_there_a_trade-off_between_peak_performance_and_performance_breadth_across_temperatures_for_aerobic_scorpions_in_teleost_fishes_/3843624.

Author's Response to Decision Letter for (RSOS-210646.R1)

See Appendix B.

Decision letter (RSOS-210646.R2)

Dear Mr Srivastava,

I am pleased to inform you that your manuscript entitled "Non-biodegradable objects may boost microbial growth in water bodies by harnessing bubbles." is now accepted for publication in Royal Society Open Science.

on behalf of Professor Roland Bouffanais (Associate Editor) and Pietro Cicuta (Subject Editor)
openscience@royalsociety.org

Appendix A

Reviewer 1

We sincerely express our gratitude to the reviewer for providing constructive feedback for our work that helped us to improve the quality of the manuscript. We have sought to address the point raised by the referee with utmost care. We believe that the manuscript is improved as a consequence.

Below is a point-by-point response to referee's concerns and suggestions. The comments from the reviewer is presented in **blue colored text** and the corresponding responses from the authors are provided as **black colored texts**. In the main manuscript and the supplementary document, the revised parts are highlighted with **yellow color**.

Using a yeast fermentation setup, the authors study, for a model waste particle (rubber), the impact of the bubble-induced waste particle motion. By measuring the yeast cell concentration and the local velocities of the rising/sinking waste particle (which belongs to the relevant results), the authors are able to quantitatively access the enhancement of mixing, sediment suspension and glucose consumption. The results are substantiated by one-dimensional modeling which is in reasonable agreement. The paper is well-written and contains results which are relevant for a variety of systems, from brewery tanks towards marine systems.

The authors would like to thank the reviewer for the positive comments.

p. 4: The numbering (a-c) in the text is somehow misleading because it does not correspond to the (a-c) in Fig.1.

Thank you for pointing it out. We, respectively, changed: (a), (b) and (c) to (i), (ii) and (iii), to avoid any misunderstanding with (a-c) of Fig.1.

Fig. 3a) and throughout the paper: An information on the size distribution of the bubbles produced, which rise and/or attach to the waste particle is missing.

Thank you for this important suggestion. The bubble size on a rising-sinking silicone rubber object follows a probability distribution like the following. The graph illustrates the probability that a particular bubble on the surface of an R-S object has a given diameter. Note that the graph peaks in the range 300-400 μm . Furthermore, the bubbles have a mean diameter of 394.7 μm with a standard deviation of 190.9 μm . We included this information in the main manuscript in the **Section 2**. Also, the shown graph is included in the supplementary document in **Section S1.2**.

Even though Figure R1 shows the wide range of bubble size distribution, the important quantity in our modelling is R-S frequency f_o and not the bubble size distribution. Because the R-S frequency f_o is used to estimate the enhanced vertical diffusivity \mathcal{D}_o due to the R-S cycle of a cuboid. We directly measure R-S frequency f_o and use it to get \mathcal{D}_o , so the present results are unaffected by the bubble size distribution.

Figure R1. The size distribution of bubbles on a rising-sinking silicone rubber object. Here N indicates the number of bubbles observed.

p. 4 what is a “standard YPD medium?”

Thank you for the question. In this manuscript, we called the standard laboratory growth medium for yeasts as the “standard YPD medium”. The composition of such medium includes 1% yeast extract, 2% D-glucose and 2% peptone in distilled water. The percentages are given as the weight of the ingredient in grams (g) divided by the volume of distilled water in milliliters (100mL). For better clarity, we included above explanation in **Section 6.1** of the main manuscript. We also added a corresponding reference on culturing yeasts as follows. Pages 11-12 of the reference may be referred for further details.

Sherman F. [1] Getting started with yeast. *Methods in enzymology*. 1991 Jan 1;194:3-21.

p.5, line 23: The term “diffusion coefficient” is not fully correct as it is the combination of advection and diffusion which matters. At least it should be renamed to “apparent diffusion coefficient”

Thank you. We agree with reviewer’s suggestion regarding the renaming of the diffusion coefficient. As advised, we fixed our nomenclature and replaced the term ‘diffusion coefficient’ with ‘apparent diffusion coefficient’ in the main manuscript.

Fig. 2a: The authors show a spike-like behavior of the R-S frequency f_0 . It remains unclear how many measured values are hidden exactly behind this curve. Thus it would be much more transparent if the measured values are given as discrete symbols instead of a solid curve. The same holds true for the bubble flux.

Thank you for the comment. To clarify the data behind the curve, as suggested, we marked the data points with discrete symbols. As a consequence, now **Fig. 2(a)** looks like the following. For the sake of tidiness, the error bars for bubble flux are omitted. However, interested readers can access the corresponding underlying data in the figshare repository through

<https://doi.org/10.6084/m9.figshare.14339027.v2>

Figure R2. The long time record of R-S frequency f_0 for a rubber cuboid along with the bubble flux f_b from the flask floor. Notice that the rise-sink only occurs during periods of weak bubbling. Each symbol displayed in the above graph is a mean of five observations. Error bars denote S.D.

p. 5: The bubble flux f_b is a somehow problematic quantity as it does not take into account the size distribution of the bubbles. In flotation, generally the bubble surface area flux is considered instead. Larger bubbles, upon attachment to the waste particle, are more effective in reducing the effective density.

Thank you for the comment. The bubble flux f_b represents the number of bubbles crossing the unit horizontal cross-sectional area (cm^2) of the flask per second. These bubbles are generated in the yeast cell layer at the flask floor. We will now illustrate a size distribution of these bubbles in following graphs:

Figure R3. (a) The size distribution of bubbles generated at the flask floor. (b) The time variation in mean bubble diameter generated at the floor. To construct this curve, 50 bubbles each were measured from $t = 20$ h, $t = 30$ h and $t = 50$ h. Error bars denote S.D.

Note from **Fig. R3(a)** that the size distribution with respect to bubble diameter is narrow with a mean value of $306.14 \mu\text{m}$ and a standard deviation of $47.5 \mu\text{m}$. This implies that the bubbles which comprise the bubble flux do not differ significantly in their sizes. To concretize it a bit more, we measured bubble sizes during the periods of weak bubbling and strong bubbling and observed that the mean bubble diameter more or less remains the same regardless of the fermentation regimes. See **Fig. R3(b)**.

This can be understood if we take into account the detachment condition of a bubble from a cavity in yeast cell layer. A bubble will detach only if the surface tension at the foot of the bubble is balanced exactly by the buoyancy force. The volume of the detached bubbled depends on the Bond number which is turn, is a function of the cavity size. In our case, since the bubble detachment diameter remains the same throughout the course of fermentation, it implies that cavity sizes in the yeast cell layer do not change with time.

More details on bubble detachment physics can be accessed through the following reference:

Lesage FJ, Marois F. Experimental and numerical analysis of quasi-static bubble size and shape characteristics at detachment. *International Journal of Heat and Mass Transfer*. 2013 Sep 1;64:53-69.

As far the bubble surface area flux S_b is concerned, it is traditionally defined as $S_b = \frac{6J_G}{d_b}$, where J_G is the superficial gas velocity and d_b is the mean bubble diameter. In our case, $J_G = f_b V_b$ such that V_b is the mean volume of a single bubble, yielding $S_b = (\pi d_b^2) f_b$. As previously shown that bubbles generated in

the yeast cell layer does not change much in our setting, therefore, the bubble flux f_b remains the only parameter of significance. The details on bubble surface area flux were referred to from the following reference

Sarhan AR, Naser J, Brooks G. CFD model simulation of bubble surface area flux in flotation column reactor in presence of minerals. International Journal of Mining Science and Technology. 2018 Nov 1;28(6):999-1007.

The information on size distribution is added in the main manuscript in **Section 2.1** and also the graphs shown above are included in supplementary document in **Section S1.3**

p.5, line 43/44: Sentence containing "...from 22 h... the effective density... is less than that of the ambient fluid,... Hence, $f_0=0$ " remains unclear. If the effective density is less than that of the ambient fluid, the object experience buoyancy and should rise. Why f_0 is then zero?

Thank you for pointing out this confusion. In our experiments, in order to calculate the R-S frequency f_o , we count the number of 'rise events that are followed by a sink event', in a ten minutes window. As in the case from $t = 22$ h to $t = 50$ h, even though the object rises to the free surface at about 22 hours, it does not sink until 50 hours. For this reason, we said that the R-S frequency is zero in between the two spikes of f_o shown in **Fig. R2**.

We added a clearer explanation of our R-S frequency measurement methods in **Section 6.3** of the main manuscript and also clarified it in **Section 2.1** to avoid any misunderstandings.

p. 7, line 50: If "c" means a number density, a renaming to N (or n) would be appropriate to avoid confusion with a concentration which typically is given as "c".

Thank you for this suggestion. The confusion is understandable. Throughout the main manuscript and the supplementary document, we changed the yeast cell concentration [cells/mL] which previously was " n " to " C_y ". Also, in **Section 3.2**, the parameters " c_w " and " c_a ", which, have the unit of [cells] are now represented as " N_w " and " N_a " respectively.

p. 8, line 6: "\delta n" is here referred to as concentration increment.... Thus, handling of n and c is confusing for me.

As the consequence of the changes made in the previous answer: " δn " is now changed to " δC_y ".

p.8, line 35: “.concentration is large compared with yeast diameter and typical cell spacing”?? These quantities have different physical dimensions. How a concentration can be larger than a cell spacing?

Thank you for pointing out the mistake, we changed the sentence to “the length scale of the bulk motion and the length scale of the yeast concentration distribution are respectively large enough compared with the yeast diameter and the typical cell spacing. Please note that the length scales for the bulk motion and for the yeast concentration distribution are of the order of a few centimeters (i.e., height of the free surface). On the other hand, the yeast diameter is approximately 10 μm and the typical cell spacing i.e., $C_y^{-1/3}$ is 10^{-2} cm during bubbling.”

It would be good to explain the total yeast flux J_T (eq. 1) in more detail to make its relation to the derivative of n with respect to time (eq. 2) more transparent.

Thank you for pointing it out. We included the derivation for the same with better clarity in **Section 3.3**. We are also including it here for easy reference.

The total yeast flux as an advective-diffusive form of the following kind

$$\mathbf{J}_T = N_{af_b}\hat{y} - u_s C_y \hat{y} - \mathcal{D}\nabla C_y + \mathbf{V}C_y \quad , \quad (1)$$

where C_y is the yeast concentration averaged over the horizontal cross section of the flask, N_{af_b} is the advective flux due to yeast-bubble adhesion pointing vertically upwards, $u_s C_y$ is the advective flux due to yeast sedimentation pointing vertically downwards, where $u_s \hat{y}$ is the yeast sedimentation velocity and $\mathcal{D}\nabla C_y$ is the diffusive flux arising due to bubble-induced mixing and the quasi-natural mixing. Also, $\mathbf{V}C_y$ is yeast advection flux due to the fluid velocity fluctuations \mathbf{V} .

If we apply the equation of continuity to eq. (1), we get

$$\frac{\partial C_y}{\partial t} = -\nabla \cdot (N_{af_b} - u_s C_y + \mathbf{V}C_y) + \nabla \cdot (\mathcal{D}\nabla C_y) + \alpha C_y \quad , \quad (R1)$$

where α is the yeast growth rate.

Integrating eq. (R1) over the horizontal cross-section of the flask gives us

$$\frac{\partial C_y}{\partial t} = -\frac{\partial(N_{af_b} - u_s C_y)}{\partial y} - C_y \frac{\partial}{\partial y} [\iint \mathbf{V} \cdot d\mathbf{S}] + \mathcal{D} \frac{\partial^2 C_y}{\partial y^2} + \alpha C_y \quad , \quad (R2)$$

where $\mathbf{dS} = dS \hat{y}$ is the elemental horizontal cross sectional area pointing vertically upwards. Now, through conservation of mass $\iint \mathbf{V} \cdot \mathbf{dS} = 0$, implying that the fluid velocity fluctuations do not contribute to the advection of yeasts (although they cause diffusion). So finally, the governing equation for yeast concentration distribution in vertical direction becomes

$$\frac{\partial c_y}{\partial t} = -\frac{\partial(N_a f_b - u_s c_y)}{\partial y} + D \frac{\partial^2 c_y}{\partial y^2} + \alpha c_y \quad , \quad (2)$$

Reviewer 2

We sincerely express our gratitude to the reviewer for providing constructive feedback for our work that helped us to improve the quality of the manuscript. We have sought to address the point raised by the referee with utmost care. We believe that the manuscript is improved as a consequence.

Below is a point-by-point response to referee's concerns and suggestions. The comments from the reviewer is presented in **blue colored text** and the corresponding responses from the authors are provided as **black colored texts**. In the main manuscript and the supplementary document, the revised parts are highlighted with **yellow color**.

In this paper, the authors report a study on the effect of bubble-induced transport of hydrophobic objects on water bodies and the microbes. They used yeast as a source of bubbles. They found that during the weak bubbling phases with hydrophobic objects, the cell growth was enhanced by the fluid agitation due to the rise-sink cycles. Using a mathematical model, they elucidated the mechanism of the enhancement. This paper is scientifically rigorous and well-written. In my opinion, the reported results are significant since the considered situation is ubiquitous and the enhancement they found will probably be utilized in industrial fermentation processes, which is not relevant to the judgement in this case. I think that this paper basically satisfies the criteria for a publication in Royal Society Open Science. So, I recommend the publication of their work after the following minor points are clarified and corrected.

The authors would like to thank the reviewer for the positive comments.

1. If ξ in Eq. 3 is constant, c_g , which must be always positive, can be negative. Is there any dependence on c_g of ξ ?

Thank you for pointing it out. While yeasts are growing, we assume that the glucose consumption parameter ξ remains constant in both phases of its exponential growth. However, as soon as yeasts growth hit the stationary phases of no growth (i.e., 50 hours post yeast inoculation), glucose consumption comes to a halt, thereby we took $\xi = 0 \text{ g cell}^{-1}\text{s}^{-1}$ after 50 hours. This was observed experimentally too. This prevents C_g from going negative. Also, throughout our experiments glucose concentration are sufficiently large to avoid any negative values. We added this explanation in the Supplementary document in **Section S2.6**. and also, in **'Materials & methods'** section of the main manuscript.

2. It is unclear how is the cell increase caused by glucose supply to the bottom simulated. Is there any C_g dependence of eq. 2?

Thank you for the question. The effect of glucose supply to the bottom is modelled as a corresponding elongation in the 'fast exponential growth phase' by two hours (i.e., the width of the first R-S frequency spike illustrated in **Fig. 2(a)**), which then affects the cell concentration in **Eq. (2)** {**Eq. (4)** in the revised manuscript}. During the second spike, the effect of the glucose supply to the floor is negligible as yeasts have already entered the stationary phase. We added this explanation in **Section 4** of the main manuscript.

3. "The increase" in Fig 5 b is unclear. In the later stage, the red line is below the blue line. More explanation is needed.

Thank you for this comment. We fixed our explanation as below:

'The increment in cell concentration $\langle C_y \rangle_{y5}$ {previously, $\langle n \rangle_{y5}$ } can be understood through an increment in glucose supply $\langle C_g \rangle_{y5}$ to the bottom during the period of first R-S spike lasting from 20-22 hours post inoculation. Note that red curve is above the blue one in this period in **Fig. 5(b)**.'

4. In page 1, there is the phrase "A great range of animals" but yeast is not an animal. The word "animal" should be corrected.

Thank you for the comment. We changed 'animals' to 'organisms'.

5. In page 5, 5mixing is probably a misspelling.

Thank you for the comment. We changed '5mixing' to 'mixing'.

6. Fig. 3 (e) in Page 3 of Supplementary document is probably a mistake.

Thank you for the comment. We changed 'Fig. 3(e)' to 'Fig. S4(e)'.

Appendix B

We sincerely express our gratitude to the reviewers and editors for providing constructive feedback for our work that helped us to improve the quality of the manuscript. We have sought to address the point raised by the referee with utmost care. We believe that the manuscript is improved as a consequence.

Below is a point-by-point response to referee's concerns and suggestions. The comments from the reviewer is presented in **blue colored text** and the corresponding responses from the authors are provided as **black colored texts**. In the main manuscript and the supplementary document, the revised parts are highlighted with **yellow color**.

Reviewer 2

The manuscript has been well improved, and I think the referees' comments have been addressed. In my opinion, the manuscript is ready for a publication.

The authors would like to thank the reviewer for recommending the publication of our work. We are grateful for their valuable comments and insights on the manuscript.

Reviewer 1

The authors have incorporated my criticism in an appropriate manner. I recommend the publication of the manuscript...

The authors are indebted to the reviewer for recommending a publication. The comments helped us to improve our manuscript.

-) p.13, line: 18: ...“includes three major regimes, viz: (i) the yeast sediment layer, (ii) the fluid layer, and (iii) the bubble layer“: Are „regimes“ the most suited description for the layers (i-ii)?

Thank you for the comment. We agree with the referee that the word ‘regime’ might not be a suitable description for the layers. Therefore, we changed the word ‘regime’ to ‘layer’ throughout the manuscript to avoid any misunderstandings.

-) An explanation of the term “standard YPD medium”, although discussed in the response, has not found entrance into the manuscript.

Thank you for the comment. We added the explanation in **Section 6.1** of the main manuscript.

-) p.18, line 12: „where $u_{s\hat{y}}$ is the yeast sedimentation velocity“: Is the "y" a typo? Should it not read only „ U_s is the velocity...“

Thank you for pointing out the mistake. We changed ‘ $u_{s\hat{y}}$ ’ to ‘ u_s ’.